# The Silent Syndrome of Long COVID and Gaps in Scientific Knowledge: A Narrative Review

**DOI:** 10.3390/v16081256

**Published:** 2024-08-05

**Authors:** Rosilene da Silva, Antonio Carlos Rosário Vallinoto, Eduardo José Melo dos Santos

**Affiliations:** 1Laboratory of Genetics of Complex Diseases, Institute of Biological Sciences, Federal University of Pará, Belém 66075-110, Brazil; rosilenesilva2005@yahoo.com.br; 2Graduate Program in Biology of Infectious and Parasitic Agents, Federal University of Pará, Belém 66075-110, Brazil; vallinoto@ufpa.br; 3Laboratory of Virology, Institute of Biological Sciences, Federal University of Pará, Belém 66075-110, Brazil

**Keywords:** long COVID, clinical features, risk factors, associated genetic factors

## Abstract

COVID-19 is still a major public health concern, mainly due to the persistence of symptoms or the appearance of new symptoms. To date, more than 200 symptoms of long COVID (LC) have been described. The present review describes and maps its relevant clinical characteristics, pathophysiology, epidemiology, and genetic and nongenetic risk factors. Given the currently available evidence on LC, we demonstrate that there are still gaps and controversies in the diagnosis, pathophysiology, epidemiology, and detection of prognostic and predictive factors, as well as the role of the viral strain and vaccination.

## 1. Introduction

The expectation and hope that the COVID-19 virus will disappear are attenuated by irregular vaccination coverage, disparities in global access to vaccines, and vaccine hesitancy, in addition to virus transmission, which is not always blocked by vaccines (despite reducing the burden of disease) [1,2]. In this sense, long COVID (LC), characterized by persistent or new symptoms (more than 200 have been described), has become a monumental challenge for science [3,4].

At an increasing rate, the global prevalence of patients recovering from post-COVID-19 infection remains alarming [5]. Although the exact number of people living with the disease’s sequelae is uncertain, approximately 10% to 20% of people infected with SARS-CoV-2 may develop symptoms that can be diagnosed as LC [6], ranging from 2% to 86% by different estimates [7,8].

Studies have not fully elucidated the extent and progression of LC or its real long-term severity. A growing number of patients continue to experience prolonged symptoms, sequelae, and other medical complications that may last weeks to months [9,10]. These symptoms can affect several organ systems at once and/or over time, and often do not disappear [11,12].

Therefore, the following questions are the focus of our research. What is the current evidence on the pathophysiology of LC sequelae, their mechanism, and their clinical manifestations? What are the known risk factors? Are there associated genetic factors? For this purpose, the present study aimed to review, describe, and discuss LC and map its relevant clinical characteristics, putative pathophysiology, risk factors, and possible associations with genetic factors.

As a review strategy and criteria for this narrative review, initially, the major themes in LC were defined: (i) history and case definition (diagnostic); (ii) physiopathology and putative mechanisms and etiology; (iii) clinical manifestations; (iv) epidemiology; (v) risk factors; and (vi) major gaps in knowledge about LC. 

Afterward, each major theme was reviewed, identifying and presenting consolidated points, gaps, and controversies.

## 2. Long COVID—Post-COVID-19 Syndrome

### 2.1. History and Case Definition

During the first year of the pandemic, groups of patients tended to experience a trend towards prolonged illness, as indicated by the persistence of symptoms for weeks or months after infection, many of whom identified themselves as “long haulers”, which contributed to the recognition of post-acute COVID-19, a syndrome characterized by persistent symptoms and/or late or long-term complications beyond 4 weeks after symptom onset [13,14]. It was then named LC by the National Institute for Health and Care Excellence (NICE) in the United Kingdom in mid-2020. The term included continuous symptomatic patients 4 to 12 weeks post-infection and patients with the syndrome beyond 12 weeks post-infection [15]. The Centers for Disease Control and Prevention (CDC) describe the condition as sequelae that extend beyond 4 weeks after the onset of infection [16]. Moreover, the WHO proposed a clinical nomenclature to unify several existing definitions: the condition that occurs in individuals with a history of probable or confirmed infection by SARS-CoV-2, usually 3 months after the onset of COVID-19, with symptoms that last at least 2 months and cannot be explained by an alternative diagnosis; thus, an ICD-10 code corresponding to the post-COVID conditions was created: U09.9 [17]. Alternatively, Greenhalgh (2020) suggested that LC, depending on the duration of symptoms, can be divided into two stages: post-acute COVID, when symptoms extend beyond 3 weeks but less than 12 weeks, and chronic COVID, where symptoms extend beyond 12 weeks [18]. 

The following terms used to define this new post-COVID-19 phase began to appear in the literature in the second half of 2020: persistent or prolonged COVID-19, subacute COVID-19 syndrome, ongoing COVID-19, post-COVID-19 syndrome, condition post-COVID-19, post-COVID-19, long COVID, post-acute sequelae of SARS-CoV-2 infection (PASC) and others. Gradually the terms post-COVID-19 and long COVID gained prominence in the databases (Figure 1).

Diagnosis has been a challenge and a bottleneck for many specialized health services. According to a systematic review using the Cochrane method, a diagnosis of LC can be made if one of four criteria is present: symptoms that persist after the acute phase of COVID-19 or its treatment, symptoms that lead to a new health restriction, new symptoms that occur after the end of the acute phase but are understood to be a consequence of COVID-19, and worsening of a pre-existing underlying condition [20]. In this context, Raveendran (2021) [21] proposed the following duration criteria for the diagnosis of LC.
In symptomatic individuals infected with SARS-CoV-2, symptoms are present for more than 2 weeks in patients with mild disease, more than 4 weeks in patients with moderate/severe disease, and more than 6 weeks in patients with critical illness.In asymptomatic individuals infected with SARS-CoV-2, the symptoms of LC persist 2 weeks after a positive reverse transcriptase–polymerase chain reaction (RT-PCR) test; the symptoms of LC persist 1 week after a positive antibody status; the symptoms of LC persist 2 weeks after a positive chest computed tomography or chest X-ray; LC symptoms begin 2 weeks after contact with suspected or positive cases of COVID-19; and in any doubtful cases [21].

Notably, there is currently no definitive diagnosis of LC. More patients continue to experience prolonged symptoms, sequelae, and other medical complications that may last weeks to months [9,10]. These symptoms can affect several organ systems at once and/or over time and often do not go away [11,12].

### 2.2. Possible Pathophysiological Mechanisms

#### 2.2.1. Tissue Damage

As more evidence is gathered to explain the underlying pathophysiology, the timing and natural course of symptoms accompanied by interdisciplinary care remain the cornerstone of therapy [22,23]. Several possible pathophysiological explanations for the persistence of symptoms after COVID-19 have often been cited: direct viral toxicity, endothelial damage, a dysregulated immune response, hyperinflammation, hypercoagulability, and poor adaptation of angiotensin-converting enzyme 2 (ACE2) [13,24]. 

However, some experts suggest that a temporal distinction be made, necessary to differentiate the acute disease from the possible sequelae of irreversible tissue damage, with varying degrees of dysfunction and symptoms involving several possible conditions, such as post-intensive care syndrome, post-complication thrombotic or hemorrhagic complications, acute phase immune-mediated complications and/or multisystem inflammatory syndrome in children or adults [25]. 

Some authors describe LC as a multisystem disorder that presents significant diagnostic challenges because many of the symptoms associated with this disorder can be easily confused with other common conditions. The manifestation of multiple-organ dysfunction is not surprising, as the SARS-CoV-2 entry receptor ACE2 is expressed in various tissues; therefore, the clinical manifestations of LC can fluctuate over time, increasing the management challenge [3,4]. 

Organs are affected not only in survivors of critical illness [11,26,27]. The impact of prolonged symptoms was observed in cases of mild COVID-19 (patients who did not require respiratory support or intensive care), moderate-to-severe cases (hospitalized patients who ended up with SARS-CoV-2 and were discharged from hospital), and mild-to- moderate cases in outpatient clinics and even in children [23,24,28,29,30,31]. 

Additionally, these clinical manifestations may be continuous or relapsing and remitting in nature. One or more symptoms of the acute phase of COVID-19 may persist, or new symptoms may appear. The mechanism underlying the appearance of new symptoms is still unknown. Most people with LC syndrome have molecular recovery (RT-PCR-negative): in other words, LC is the illness experienced between molecular recovery and clinical recovery [32].

#### 2.2.2. Lungs

The pathophysiology of COVID-19 is similar in many respects to the pathophysiology of SARS, with aggressive inflammatory responses strongly implicated in airway damage in a substantial proportion of patients [33]. The infection can, initially, destroy lung cells and as consequence trigger a local immune response through the recruitment of macrophages and monocytes that respond to infection by promoting the release of cytokines. On the other hand, in a smaller number of patients, there is an initial atypical and systemic immune response that triggers substantial activation of cytokines (cytokine storm), causing damage to infected cells and severe lung disease. Airway epithelial cells can cause programmed cell death due to an inflammatory response (pyroptosis) [34]. 

In follow-up studies of patients who survived COVID-19, pulmonary radiological abnormalities (71%) and functional impairments (25%) were found, although less than 10% had severe pneumonia [35]. Reduced pulmonary diffusing capacity is associated with radiological abnormalities in 42% of patients 3 months after hospital discharge, regardless of initial disease severity [36]. Even 6 months after the onset of symptoms, pulmonary radiological abnormalities associated with persistent symptoms were still present in approximately half of the patients [37]. Radiological evidence of pulmonary fibrosis lasting up to 6 months after hospital discharge is also associated with the initial severity of the disease [38,39]. 

Lung function was also worse among young recruits who had symptomatic COVID-19 than among non-COVID-19 recruits, manifesting as lower aerobic capacity, at around 45 days of follow-up [40]. In addition, impaired pulmonary gas exchange function was observed among hospital-discharged patients who had moderate COVID-19 compared to healthy controls [41]. Pulmonary scarring may be a common sequela that is responsible for persistent dyspnea and cough in LC patients [39]. LC symptoms may persist even in those with improvements in radiological and functional lung exams [42].

#### 2.2.3. Brain

SARS-CoV-2 can enter the central nervous system via the nose and reach the olfactory bulb region of the brain to trigger infection [43]. Growing evidence supports the neuroinvasion potential of SARS-CoV-2, and persistent effects on the brain can be detected even in individuals who have had the mild form of the disease [44]. In agreement with the neuroinvasion, studies found that the brainstem expresses higher levels of ACE2 than other brain regions [45]. 

Thus, neurological manifestations may include central nervous system manifestations (dizziness, headache, impaired consciousness, acute cerebrovascular disease, ataxia, and convulsion), peripheral nervous system manifestations (taste deficiency, smell deficiency, vision deficiency, and neuropathic pain), and neuromuscular manifestations (skeletal and muscle damage) [43]. In this scenario, autopsies of brain tissue from nonhuman primates infected with SARS-CoV-2 showed multiple evidence of neurological damage, including in animals that did not develop severe respiratory disease, potentially providing information about the neurological manifestations of LC [46]. 

Long-term neurological symptoms seem to be more likely in patients with severe cases of COVID-19 [43]. Delirium has been reported in 20–30% of hospitalized patients [43] and is a strong predictor of long-term cognitive impairment, especially among older adults [47]. An additional study showed that patients who were admitted to the intensive care unit (ICU) were 56% more likely to develop a neuropsychiatric disorder than survivors outside the ICU. In that same study, which included 236,379 survivors of COVID-19, within 6 months after the onset of the first symptoms, about one-third of the patients received a neuropsychiatric diagnosis such as stroke, dementia, insomnia, anxiety, or mood disorders [48].

#### 2.2.4. Heart

Cardiac injury was also observed in LC. The inherent structure of SARS-CoV-2 and its interaction with ACE2-mediated pathways have been implicated in the development of cardiovascular manifestations, progressively resulting in acute respiratory distress syndrome (ARDS), multiple-organ failure, cytokine release syndrome, and myocardial damage [49]. The renin–angiotensin–aldosterone system (RAAS) may be dysregulated by ACE2 receptors in the cardiovascular system, leading to ventricular remodeling that alters myocardial demand and induces further damage to cardiac cells [50]. Systemic oxidative stress in COVID-19 induced by hypoxemia directly damages cardiac cells via mitochondrial damage and intracellular acidosis, and the local and systemic effects of the immune system damage the cardiac microvasculature, resulting in perfusion defects [51,52]. 

The cardiac manifestations of LC are believed to be caused by cardiac damage during the acute phase of COVID-19 through inflammation, which can manifest itself severely when there is an exacerbation of pre-existing cardiac complications. The effects on the heart and blood vessels can be divided into acute complications (heart failure, rhythm disturbances, weakness, and inflammation of the heart muscle) and chronic complications such as hypertension and diabetes [49]. 

A prospective cohort study analyzed the cardiovascular magnetic resonance (CMR) data of 100 patients discharged from a hospital stay for COVID-19 and found that the cardiac abnormalities and myocardial inflammation in 78% and 60% of participants, respectively, were not associated with the initial severity of COVID-19 [53]. Another study showed that radiological abnormalities in ventricular remodeling were still evident in 29% of 79 COVID-19 survivors, even 3 months after hospital discharge [54]. Cardiac symptoms such as chest pain, heart palpitations, and tachycardia often persist for up to 6 months among COVID-19 survivors, suggesting substantial cardiac sequelae [11,37]. Moreover, a single study [55] speculated that abzymes, which are antibodies with catalytic activity, could play a role in hypotension, a singular symptom observed among a discrete number of patients. 

Finally, Mauro et al. (2024) reported decreased heart rate variability at rest, measured by finger photoplethysmography, among 58 health -care workers (HCWs) with LC in Trieste University hospital—all but one previously developing mild COVID-19—assessed after a median time of 419.5 days since COVID-19 diagnosis. Slow-paced breathing was able to enhance vagal response and restore baroreflex modulation and vascular reactivity in HCWs with LC [56].

#### 2.2.5. Thrombotic Events

Thrombotic events are common in patients with COVID-19 [57] and may result from inflammatory processes in acute respiratory infections of viral origin [58]. Complications such as diffuse intravascular coagulation and thromboembolic events involving different organs are well recognized during the acute phase of the disease [57].

In the context of LC, thromboinflammatory events seem to be relevant as suggested by Pasini et al. (2021), who studied the blood serum profile of 75 patients with LC (2 months after hospital discharge). In that study, all patients had very high serum concentrations of ferritin and D-dimer [59]. Furthermore, the study of Patterson et al. (2022) highlights the persistence of the S1 spike protein in CD16^+^ monocytes for up to 15 months in patients with LC [60], which may contribute to the production of microclots as part of the mechanism of coagulation through the amplification of trigger proteins [61]. However, while D-dimer was found to increase in LC caused by all variants, fibrinaloid clots occur with more intensity in pre-Omicron variants [61]. Additionally, the rate of venous thromboembolism in the context of LC is thought to be less than 5% [57]. Larger studies of well-characterized clinical cohorts are needed due to the potential for persistent hypercoagulability in patients with LC [62,63]. 

Finally, Nicolai (2023) put forth three possible hypotheses explaining thromboinflammation in LC: lasting structural changes, especially endothelial damage, caused during the initial infection; a persistent viral reservoir; and immunopathology driven by a misguided immune system [62].

#### 2.2.6. Other Neurological Organs and Systems 

It is possible that SARS-CoV-2, through ACE2 receptors present in the conjunctiva or through the nasolacrimal duct, helps in the spread of infection to other organs of the human body [64]. Possible cognitive changes may present with ophthalmological symptoms or changes in oculomotor function, as 20% to 25% of the brain cortex is involved in visual or oculomotor processing [65]. The most common visual symptoms reported in LC are decreased visual acuity with blurred vision, difficulty moving objects or tracking objects, difficulty with changes in light, and difficulty reading [66]. 

In the upper and lower gastrointestinal tract, viral infections, as well as bacterial and protozoal infections, are risk factors for the development of functional gastrointestinal disorders like postinfectious dyspepsia and irritable bowel syndrome [67]. Although the cause of gastrointestinal symptoms in patients with LC is unknown, the proposed hypothesis is the presence of the virus itself, which infects the gastrointestinal tract, causing changes in the intestinal microbiome, systemic inflammatory response, and critical illness [68,69].

Metabolite signatures associated with LC have been investigated, and serotonin was the molecule most significantly associated with LC [70]. Serotonin (5-HT) is a molecule produced mainly in the enterochromaffin cells of the gastrointestinal epithelium and is released into the circulation, targeting enterocytes, smooth muscles, and enteric neurons [71]. Wong et al. (2023), in a cohort study on 58 patients with LC, pointed out that serotonin depletion due to viral inflammation may be caused by reduced tryptophan absorption, platelet hyperactivation and thrombocytopenia, and increased expression of monoamine oxidase (MAO) enzymes. Peripheral reduction in serotonin in turn impedes vagus nerve activity and therefore impairs hippocampal responses and memory [70]. However, a recent study suggested that although the role of peripheral 5-HT in LC remains uncertain, reduction in platelet 5-HT may be associated with a decrease in cardiovascular sequelae of COVID-19, due to effective inhibition of the platelet serotonin transporter by selective serotonin reuptake inhibitors (SSRIs), and therefore a sharp decline in platelet serotonin is expected, not necessarily associated with cognitive symptoms [72].

#### 2.2.7. Gastrointestinal Organs and Systems 

The composition of the gut microbiota is significantly altered in patients with COVID-19 [73]. The gastrointestinal tract is the body’s largest immune organ, and its resident microbiota is known to modulate host immune responses [74]. Yeoh et al. (2021) in a cohort study, showed that the intestinal microbiome is involved in the magnitude of the severity of COVID-19, possibly through the modulation of host immune responses, and may contribute to the persistence of symptoms [75]. Additional studies also showed that the microbiota of different anatomical areas is altered in patients with LC and the possible existence of links between dysbiosis and susceptibility and/or severity of LC [75,76,77]. However, the perception of gastrointestinal symptoms is very subjective under stressful situations like pandemics, and it is important to take this into account during the evaluation of gastrointestinal LC signs and symptoms [68,69]. Gastrointestinal symptoms may persist and suggest a worsening of COVID-19. Noviello et al. (2021) showed that gastroenterological symptoms are associated with a greater risk of chronic fatigue and somatoform disorders (extraintestinal symptoms such as headache and joint and muscle pain, which, in the absence of organic/biological changes that explain them, are known as functional somatic syndromes). In the same study, mild gastroenterological symptoms persisted 5 months after infection, particularly in patients who reported diarrhea in the acute phase [67]. A cohort study showed that 5% of patients reported diarrhea or vomiting 6 months after infection with SARS-CoV-2 [37]. 

Moreover, Álvarez-Santacruz et al. (2024) in a study on the microbiota of the upper respiratory tract (including the oral, nasal, oropharyngeal, and mainly nasopharyngeal microbiota) highlights the relevant changes in bacterial abundance identified in the oral microbiota of patients with LC that may be inducing persistent inflammation, thus suggesting, that the microbiota of the upper respiratory tract may contribute to the pathogenesis of LC [76].

Multiple factors can lead to liver injury, including the direct cytopathic effect of the virus through lysis or induction of apoptosis, inflammation, intrahepatic immune activation, microvascular thrombosis, hepatic congestion, interruption of the liver-intestinal axis, drug toxicity and multidrug interactions [78]. The expression of the ACE2 receptor in bile duct epithelial cells (57.7% cholangiocytes) is markedly higher than that in liver cells (2.6% hepatocytes) [68]. Mitochondrial swelling, endoplasmic reticulum dilation, decreased glycogen granules, and cell membrane dysfunction in hepatocytes, in addition to massive hepatic apoptosis, have been reported in COVID-19 patients [79]. 

Concerns about long-term direct liver damage have grown since the emergence of cholangiopathy and persistent cholestasis in LC, and individuals with chronic liver disease and cirrhosis are particularly susceptible to adverse outcomes [80].

#### 2.2.8. Other Organs and Systems 

The kidney seems to be a vulnerable organ in patients with COVID-19, with evidence of acute kidney injury (AKI) in 19.45% of patients and a significantly increased risk of in-hospital mortality (54.2%) [81]. The entry mechanisms of SARS-CoV-2 include ACE2 receptors and transmembrane serine protease 2 (TMPRSS2) in renal cells [81]. The causes of AKI include hemodynamic changes, hypovolemia, viral infection directly causing renal tubular injury, thrombotic vascular processes, glomerular pathology, and rhabdomyolysis, and may be associated with hematuria, proteinuria, and abnormal levels of serum electrolytes (for example, potassium and sodium) [82]. In patients with LC, AKI is characterized by renal manifestations such as a decreased glomerular filtration rate and hematuria, especially in patients who have severe disease [63]. 

Sequelae that affect the endocrine system have been described. The course of the pandemic has shown that obesity, diabetes, and metabolic syndrome are among the main risk factors for the development of severe disease and that they have an increased risk of prolonged symptoms in LC patients [24,37]. There are reports that SARS-CoV-2 infection can induce or worsen diabetes mellitus (type 1 and 2), considering that infection and inflammation drive diabetes, making it a risk factor [83]. This can be attributed to the increase in glycolytic activity in inflammatory cells, such as macrophages, resulting in more epithelial damage by exacerbation of the viral load and increased production of proinflammatory cytokines and subsequent T-cell dysfunction and lung epithelial cell death [84]. In addition, other coronaviruses are known to cause damage to pancreatic islet cells (due to the expression of ACE2) and result in acute type 1 diabetes mellitus [85]. New-onset diabetes and diabetic ketoacidosis have been reported in LC patients with persistent glycemic abnormalities [63]. 

Multiple cutaneous manifestations are associated with acute COVID-19, including hair loss, rash, erythema pernio, and acral skin lesions (cutaneous manifestations associated with systemic diseases) [86]. Their pathophysiology is controversial. The presence of SARS-CoV-2 in skin samples was related to the expression of ACE2 in the endothelial cells of cutaneous capillaries [87]. It has been hypothesized that the virus could cause the reactivation of other latent viruses, such as human herpesvirus 6 [88]. The literature indicates that the manifestations may present clinical characteristics like those of viral exanthems and skin rashes secondary to the systemic consequences of COVID-19, such as vasculitis and thrombotic vasculopathy [89]. 

Additionally, acute telogen effluvium, a cause of diffuse hair loss that occurs after a triggering event such as an inflammatory state, and appears and resolves in 3 to 6 months, has been reported in patients recovering from the infection, stress being a putative contributing factor [90]. Rash and persistent hair loss have been reported in cohort studies on LC [37].

### 2.3. Inflammation and Immune Response 

The sequelae of COVID-19 are believed to result from low-grade chronic inflammation triggered by the immunomodulatory effects of the SARS-CoV-2 virus. This virus, through the airways, reaches the lung epithelium, through which viral particles can invade the bloodstream, causing a peak in viremia and hematogenous dissemination in heart-striated muscle, endocrine glands, and any other cells that have ACE2 on their surface [91].

Regarding the humoral response, the accumulation of memory cells and secretory cells (subpopulations of B cells that express antibodies) becomes essential for controlling the persistent phase of infection. B cells respond to both SARS-CoV-2 protein N and protein S through monoclonal antibodies (mAbs) after the onset of symptoms; in MERS-CoV in vitro and in MERS-CoV-infected monkeys, the human monoclonal antibody m336 exhibits high neutralization activity, with a reduced viral load [92]. 

SARS-CoV-2 stimulates specialized cells to present antigens to self-reactive T cells in a process called bystander activation, similar to what happens in autoimmune diseases [93]. Thyroid dysfunction, for example, may play a role in the pathophysiology of LC autoimmunity, as the thyroid is closely linked to T-cell-mediated autoimmunity [94]. 

Moreover, B cells may also be involved in LC autoimmunity, as shown by Zuo et al. (2020), who found that antiphospholipid autoantibodies were associated with neutrophil hyperactivity and more severe clinical outcomes (detected in 52% of serum samples from patients hospitalized with COVID-19) [95]. Another study also identified autoantibodies against interferons, neutrophils, connective tissues, cyclic citrullinated peptides, and cell nuclei in 10–50% of patients with COVID-19 [96].

Two additional studies suggested the role of inflammatory response in long-lasting symptoms. The first was a study of 18 cancer patients (with solid tumors or hematological malignancies) who had neurological sequelae of COVID-19, which detected inflammatory cytokines in the leptomeninges associated with the presence of neurological symptoms 2 months after infection, and the levels of matrix metalloproteinase 10 (a protein involved in the breakdown of the extracellular matrix in disease processes such as arthritis and metastasis) in the spinal fluid correlated with the degree of neurological dysfunction [97]. The second one found that patients with LC syndrome may develop a dysfunctional immune response, with increased IFN-γ, IL-2, B cells, and CD4^+^ and CD8^+^ T cells, and seem to have activated effector T cells with proinflammatory characteristics. Some patients may also have an inadequate innate response to interferons and/or macrophage activity and even a genetic predisposition [98].

### 2.4. Clinical Manifestations 

Given the broad spectrum of clinical manifestations, the etiopathogenesis of LC syndrome is likely multifactorial [98]. The spectrum of symptoms is systemic, and the pathogenesis may be related to the direct action of SARS-CoV-2 or a consequence of the thromboembolic inflammatory response to infection [9]. Symptoms may present in a multifaceted, heterogeneous, and relapsing–remitting manner [99]. Several organs and systems can be affected, such as respiratory, cardiovascular, neurological, dermatological, musculoarticular, endocrine, renal, and hepatic. The literature describes more than 200 symptoms or sequelae [3,4]. Prolonged symptoms are common after many viral and bacterial infections, but although clinical features have also been observed after influenza infection, the incidence of them after COVID-19 seems to be higher [48]. 

Evidence points to mitochondrial dysfunction as a potential underlying mechanism contributing to the persistence and constellation of LC symptoms [100]. A longitudinal case–control study by Appelman et al. (2024) revealed that patients with LC worsened after exercise induction, skeletal muscle structure was associated with lower exercise capacity and local and systemic metabolic disorders, intense exercise-induced myopathy and tissue infiltration of amyloid-containing deposits in skeletal muscles [101]. Limited exercise tolerance and post-exertional malaise in LC have been described [74], and possible factors may include mitochondrial dysfunction, systemic inflammation, forced physical inactivity or disuse, viral infiltration, hypoxemia, malnutrition, and certain medications [102]. Mitochondrial function in skeletal muscle is a critical determinant of systemic fuel homeostasis, and its impairment can lead to decreased energy production, increased production of reactive oxygen species, and the initiation of inflammatory pathways [102,103].

Both the duration and intensity of LC symptoms and signs vary widely [104]. Thus, regarding symptom duration, Nehme et al. (2022) using logistic regression in a large-cohort study (767 subjects), considering symptom chronicization as the continuous presence of symptoms at each evaluation time point (at 7 and 15 months) [104]. Overall, 47.9% of the patients experienced symptom resolution at the second follow-up (15 months after infection), 52.1% of whom experienced persistent symptoms and were considered to have a chronic condition [104]. However, there is heterogeneity in the literature regarding the proportion of patients whose symptoms resolved. For example, while Sansone et al. (2023), who followed 247 patients, found that 75% of them had symptoms resolved before an average of 15 months [105], Huang C et al. (2022) followed 1190 patients for 2 years and reported that 45% had the symptoms of LC resolved with success [106]. This observed heterogeneity may be intrinsic to LC or due to some type of bias, such as that introduced by difficulties in diagnosis, as discussed previously. The same problems in diagnosis may also be related to the high variation in LC prevalence, also reported in Table 1.

Regarding the association with chronicity, the results showed an independent association between difficulty concentrating at 7 months and the chronicization of symptoms [104]. Another study showed that at 7 months, many patients continued to experience a significant increase in symptoms and had not yet recovered (mainly from systemic and neurological/cognitive symptoms) [3]. 

The most common manifestations are systemic (fatigue and lack of concentration), neuropsychiatric (sleep abnormalities, chronic headache, “brain fog”, memory and mood impairment, and pain syndromes), cardiac (palpitations, syncope, dysrhythmias, and posture), and respiratory (dyspnea and cough) [63]. “Brain fog” is a term used by patients to define the escape of words, forgetfulness, and persistent cognitive difficulties in LC [3]. 

Since 2020, several publications (cohort, case–control, and cross-sectional studies) had catalogued the main clinical manifestations of LC, highlighting the symptoms most common and the prevalence and duration of symptoms. 

Thus, we conducted a direct search for a set of articles that addressed the research topic of LC syndrome and original, clinical, and follow-up studies. After a thorough analysis, we chose a total of 50 articles to show the main clinical manifestations presented by the patients (Figure 2). The following characteristics were extracted from the studies: first author, year, country, sample, sex, severity, duration of symptoms, prevalence (percentage of patients with the syndrome), and most common symptoms ordered by frequency above 20% (Table 1).

Figure 2 shows the main symptoms cited in the studies presented in Table 1 according to the frequency of citations, mean frequency of symptoms, and percentage of manifestations in the various body systems. Fatigue, dyspnea, and amnesia (70%, 64%, and 24%, respectively) were the most identified symptoms. On average, of the symptoms, sensorimotor symptoms, fatigue, and sore throat (60%, 51%, and 49%, respectively) had the highest averages.

The complexity and number of LC symptoms have a significant impact on quality of life, as demonstrated by our group [144] and other studies [106,123,145,146]. The influence of this condition on work ability stands out, as highlighted by Sansone et al. (2023) and measured by the Work Ability Index [105]. These studies point to the socioeconomic and psychological consequences of LC that need to be better understood, allowing for improved management.

### 2.5. Epidemiology 

#### 2.5.1. Long Covid Prevalences Worldwide

Globally, the number of patients with post-COVID-19 sequelae continues to increase at an unprecedented rate [5]. Although the exact number of people living with sequelae of the disease is uncertain, some 10–20% of people infected with SARS-CoV-2 may be diagnosed with LC [6]. This percentage ranges from 2% to 86% in different studies [7,8]. A systematic review with meta-analysis showed that the estimated global combined prevalence of LC was 43%, with estimates of regional prevalence of 51% in Asia, 44% in Europe, and 31% in North America [147]. In Africa, a study by Müller et al. (2023) revealed wide variation in the prevalence of LC, ranging from 2% in Ghana to 86% in Egypt [8].Another study including four African countries reported that the prevalence of post-COVID-19 conditions among health professionals was 58.8% [148]. A comparison of the prevalence of one or more symptoms in a hospitalized population on all continents showed that the pooled prevalence was greater in Europe than in Asia, North America, and elsewhere (Africa and Oceania), at 62.7%, 40.9%, 38.9%, and 24.8%, respectively [149]. According to a review by Chen et al. (2022), 54% and 34% of hospitalized and nonhospitalized patients, respectively, had LC [147]. 

Prolonged disease may occur among young adults without underlying comorbidities and in patients who have had mild disease [22]. Approximately 12% to 15% of patients who have mild symptoms remain symptomatic for up to 8 months [150]. Among non–hospitalized patients, the risk of LC increased with the number of symptoms during the acute disease and with pre-existing comorbidities [151]. 

Sexual dimorphism was suggested by Bechmann et al. (2022), who highlighted the difference in the severity and survival of COVID-19 between males and females infected with SARS-CoV-2. Evidence suggests that the severity and mortality of COVID-19 are greater in males than in females, whereas females seem to have a greater risk of reinfection and the development of LC [17,104,152]. 

#### 2.5.2. Long Covid Symptoms Prevalences

Regarding the prevalence of symptoms, approximately 63% of the patients reported at least one symptom after 30 days of symptom onset/hospitalization, 71% reported at least one symptom after 60 days, and 46% reported at least one at 90 days [153]. In another study, 54% of the patients reported at least one symptom at 1 month, 55% reported at least one symptom at 2–5 months, and 54% reported at least one symptom at 6 months or more [154]. Other studies have reported much lower rates of continuous symptoms after 12 weeks (2.3% to 3%) [10]. 

Regarding symptoms duration Huang et al. (2022) followed up patients for up to 2 years, and persistent symptoms were evident in a significant number of patients, fatigue and muscle weakness being the most frequent, in addition to problems of pain, discomfort, anxiety, and depression. These differences at 2 years were more evident in LC patients than in controls [106]. Nevertheless, most people have good functional recovery during 1 year of follow-up [155]. 

Evidence from cross-sectional, cohort, and case-control studies suggests that persistent physical symptoms after COVID-19, particularly fatigue, are most common [10,48,156]. A systematic review with meta-analysis indicated that the five most common symptoms were fatigue (58%), headache (44%), attention disorder (27%), hair loss (25%), and dyspnea (24%) [9]. Moreover, in children and adolescents, estimates of the prevalence of LC range from 1% to 51%, and 14% have ongoing symptoms [31]. The most prevalent symptoms are mood swings, fatigue, and sleep disorders [157] or multiple symptoms (tiredness, headache, and shortness of breath) in 29.6% of patients [31].

In Brazil, despite the scarcity of data, LC has been reported. Clinically significant symptoms, including fatigue, dyspnea, cough, headache, and muscle weakness, loss of smell or taste were the most prevalent [158,159,160] in agreement with most worldwide studies.

### 2.6. Risk Factors

LC has become a complex and heterogeneous syndrome that may have multiple triggering factors. According to Sudre et al. (2021), these factors include disease severity, age (over 50 years), sex (female), and pre-existing comorbidities, such as asthma or respiratory disease, obesity, and high body mass index. Sudre et al. (2021) also showed a higher risk of persistent LC when they had more than five symptoms during the first week of COVID-19 (odds ratio = 3.53; confidence interval: 2.76–4.5). Furthermore, this study shows that 13.3% of participants had symptoms for more than 28 days, 4.5% had symptoms for more than 8 weeks, and 2.3% had symptoms for more than 12 weeks. Chronic symptoms affect 10% of individuals aged 18 to 49 years, but the proportion increased to 22% in individuals older than 70 years [10]. 

According to Yong S (2021), the risk factors for LC are not yet well established, but they seem to include female sex, more than five initial symptoms of acute infection, previous dyspnea, and psychiatric disorders [23]. Other studies also indicated that older or younger age, female sex, obesity, smoking status, severe clinical conditions, and a greater number of symptoms were potential risk factors [7,22,48,152,161,162,163].

Interestingly, female sex has been associated with lesser severity of COVID-19 making these epidemiological data in disagreement with other studies if contrasted, for example, with Cegolon et al. (2023) who found associations of LC with more severe acute disease, viral shedding time during pre-Omicron strain waves, being the risk lower after Omicron, reinfections, and humoral immunity [164].

Moreover, regarding the impact of SARS-CoV-2 variants, a case–control study showed that the risk of developing LC may be lower among people infected with the Omicron variant (4.5%) at all vaccination times than among those infected with the Delta variant (10.8%) [165]. Another case-control study with 7051 health professionals showed that having two or more SARS-CoV-2 infections, female sex, and older age put people at higher risk of developing LC. Otherwise, infections with Delta or Omicron strains, as well as more than 4 vaccine doses seemed to be protective against LC [130].

Therefore, nonvaccination may also be a risk factor for LC. In a systematic review, all studies showed that vaccines reduced the risk of contracting mild-to-moderate COVID-19, supporting the hypothesis that vaccination could be used as a preventive strategy to reduce long-term symptoms. However, most of these studies included infected patients 1 week to 1 month after vaccination, a short period [166]. A study in the United Kingdom involving more than 28,000 patients showed a 12.8% reduction in the chances of developing LC after the first dose of the vaccine and an additional 8.8% reduction after the second dose [167].

Age may also be related to the risk of LC. According to Koc et al. (2022), older individuals are more likely to have a greater number of morbidities and tend to develop more severe forms of COVID-19, contributing to the development of LC [168]. Considering participants aged between 18 and 70 years, Thompson et al. (2022) showed that the risk of developing LC symptoms increased with age. Interestingly, a decreased risk of LC was observed in people older than 70 years, which may be a result of competitive mortality, nonresponse bias, and/or incorrect attribution of LC to another condition [169]. 

There is evidence that COVID-19 during pregnancy substantially increases the risk of pre-eclampsia [170], which may increase cardiovascular disease later in life [171], but the impact of LC on pregnant females and postpartum females is unclear [172]. Machado and Ayuk (2023) found that pregnancy did not seem to increase the risk of developing LC, but persistent symptoms of pregnancy were related, such as fatigue, shortness of breath, and mental confusion, and the physiological changes after childbirth can lead to a change in the nature and severity of these symptoms [173]. A cohort study followed live births among SARS-CoV-2-positive mothers for 12 months and showed an association between maternal positivity and a higher rate of neurodevelopmental diagnoses in some children (6.3%) [174]. 

Regarding comorbidities as risk factors, although diabetes, hypertension, and hypercholesterolemia are risk factors for severe disease and mortality in the acute phase of COVID-19, in LC, there is no strong evidence of associations. Asthma was the only specific medical condition associated with an increased likelihood of having symptoms for more than 4 weeks [169]. Moreover, Cancer and immunosuppression are risk factors for the severity and mortality of COVID-19, but there is no evidence of their association with LC [169]. Considering that chronic inflammation increases the risk of oncogenesis, the development of cancer may be one of the predictable sequelae of COVID-19 [175]. There is a hypothesis that SARS-CoV-2 (as with many oncogenic viruses) can cause cancer in different organs using different strategies [176]. The literature has shown an association between viral infections and various types of cancer, as well as chronic inflammation and immune escape in oncogenic transformation [177]. According to Saini and Aneja (2021), the development of cancer is a consequence of long-lasting mutagenic events combined with other carcinogenic events, and COVID-19 can predispose the body to the development of cancer and accelerate its progression [175]. 

In addition, in studies on the direct and indirect impacts of the COVID-19 pandemic on the treatment of lung cancer patients, all the results consistently confirmed that lung cancer predicts a worse outcome in SARS-CoV-2 infection [178]. Cancer survivors with LC have cardiovascular, pulmonary, sensory, and musculoskeletal manifestations [13]. Symptoms such as nausea and loss of smell or taste among patients with LC can also cause significant impairment in lifestyle and quality of life and may be associated with a lack of appetite and weight loss, symptoms that are not uncommon among long-term survivors of head and neck cancer [179]. Regarding mental health, prepandemic psychological distress was associated with a greater risk of LC outcomes for more than 12 weeks [169].

### 2.7. Association with Cellular, Molecular, and Genetic Factors

Regarding changes in plasma protein level and cellular changes, which are strongly associated with genetic factors, some studies have pointed to an important role of host genetics in LC. Immunological signatures associated with symptoms gradually increase. In a cohort of 309 patients, the global proteomic and metabolomic profiles of plasma were investigated to identify plasma markers of LC in patients with up to 3 months of symptoms associated with different types of sequelae. Immunological associations were observed between factors that contributed to the sequelae of LC, which appear to decrease over time, evidencing different immunological states of convalescence [180]. In this study, the reactivation of latent viruses during the initial infection seemed to contribute to LC sequelae, which was also observed after the acute expansion of cytotoxic T cells in patients with gastrointestinal sequelae. Furthermore, in the plasma of patients in the acute stage of the disease, multiple inflammatory biomarkers, including IFN-γ, C-reactive protein (CRP), and IL-6, which are positively associated with autoantibodies, were detected in patients with LC [180]. 

Changes in proinflammatory biomarkers have been reported in LC. Two months after hospitalization for COVID-19, lung lesions were associated with elevated systemic inflammatory biomarkers, such as d–dimer, IL-6, and CRP. Multivariate regression showed that the IL-6 level at admission was an independent factor associated with persistent lung injury [181]. Another study found that 3 months after hospital discharge, COVID-19 survivors had high levels of blood urea nitrogen and d–dimer, which are considered risk factors for pulmonary dysfunction [35]. Among LC patients followed for 3 months, Liang et al. (2020) reported that lymphopenia correlated with chest tightness and heart palpitations, while elevated troponinI correlated with fatigue [129]. 

However, inflammatory biomarkers can fluctuate in autoimmune diseases and other chronic inflammatory diseases depending on disease activity and patient characteristics [182]. 

Our group also contributed to this field. A possible molecular signature for LC was suggested by Queiroz et al. (2022), characterized by an inflammatory profile of T-helper 17 cells with reduced anti-inflammatory response and reduced levels of the anti-inflammatory cytokines IL-4 and IL-10 [183]. Another observational study by our group showed that the increase in temperature and the reduction in heart rate variability were directly related to the increase in inflammatory cytokines and the reduction of anti-inflammatory cytokines. Cytokine analysis revealed increased levels of IL-17 and IL-2 and decreased IL-4, suggesting a reduction in parasympathetic activation during LC and an increase in body temperature due to possible endothelial damage caused by the maintenance of high levels of inflammatory mediators [184]. 

Regarding genetic factors preliminary results of a genome-wide association study published in preprint showed a single signal in the region of the *FOXP4* gene, which appears to be associated with lung dysfunction and severity of COVID-19, as well as the development of LC [185]. Additionally, Da Silva et al. (2023) analyzed the roles of 10 functional polymorphisms, chosen based on their functional characteristics, most of which are related to the modulation of gene expression in genes that encode cytokines involved in the main inflammatory pathways of COVID-19 and proteins associated with the risk of thrombophilia, as well as other single-nucleotide polymorphisms (SNPs) associated with the regulation of human leukocyte antigen (HLA) receptor expression antigen class II and dendritic cells. The findings showed that the proportions of symptoms in the acute phase of the disease were higher among patients who progressed to the clinical picture of LC. The AA genotype of the interferon-gamma (*IFNG*) SNP rs2430561 was more frequent among patients with LC and was associated with the presence of certain symptoms. The CC genotype of the methylenetetrahydrofolate reductase (*MTHFR*) SNP rs1801133, which is associated with thrombophilia, was also more frequent among patients with LC [186].

More recently, Queiroz et al. (2024) reported an association between LC and high expression of the cyclic GMP-AMP synthase (cGAS), stimulator of interferon genes (*STING*), and interferon-alpha (*IFN-α*) genes. Activation of the cGAS-*STING* pathway may contribute to LC due to autoinflammatory disease in some tissues after resolution of infection in severe COVID-19 [187].

## 3. Conclusions and Future Prospects

This review is narrative. Thus, as an important limitation, the selection criteria for studies to be included and cited are more subjective than those for systematic reviews. However, the literature on each main subtheme of LC has been extensively researched and points of convergence or agreement have been consolidated and represented by main publications. Furthermore, gaps in knowledge were identified, as well as some controversies. Some themes, such as the enormous heterogeneity in the prevalence of LC, become linked to the difficulty of diagnosis and other issues of underreporting, highlighting the strength of a general narrative review that allows integrated views between the subthemes of LC.

This review summarizes the currently available evidence on LC. Most patients who have recovered from COVID-19 have symptoms and postinfection sequelae, which may persist for a long time. The length of time that symptoms and sequelae last, as well as the time needed for recovery, are still questions that need investigating. 

Much is still not understood about the pathophysiology and detection of prognostic and predictive factors: the factors underlying the development of and recovery from LC are poorly understood. The sequelae acquired after infection are diverse and directly impact people’s quality of life. Even after greater control of COVID-19 through vaccines, the disease remains a public health challenge, and there is a clear need for a better and more homogeneous profile of LC, allowing reproducibility of the detection of putative risk factors. Thus, it would be desirable to detect regional differences and similarities in the patterns of LC, allowing the evaluation of the influence of environmental and genetic specificities of the host, such as by addressing nutritional status, quality of life, socioeconomic factors, the genetic profile of the host, and patterns of coinfection. Moreover, the diagnosis of LC is still presumptive and circumstantial, compromising the epidemiological and etiological understanding of the disease. Additional dimensions, such as the role of such factors as viral strain, multiple infections, vaccination, host genetics, and environmental factors and specificities, require an accumulation of evidence that would provide clues and suggest directions for better-controlled studies.

## Figures and Tables

**Figure 1 viruses-16-01256-f001:**
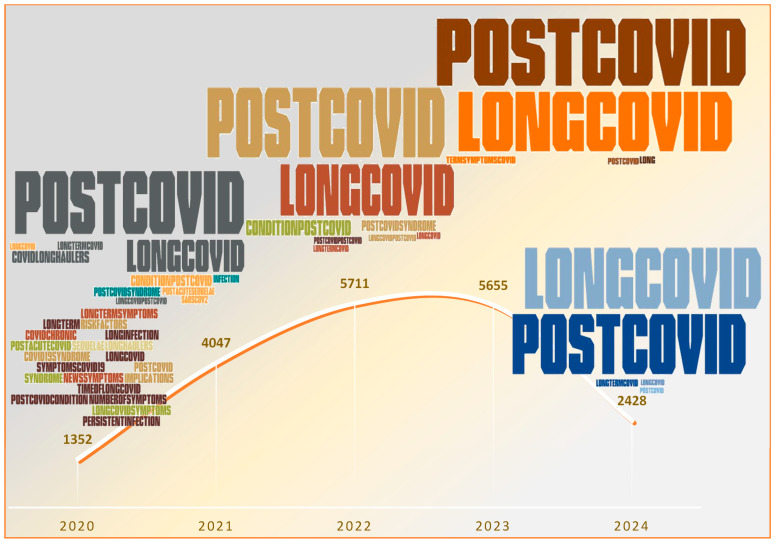
Terms used to define long-term COVID-19 over the period 2020–2024. In the timeline, the font size is proportional to the percentage of publications that use each synonym. The total number of publications using any of these terms is also presented. The search was performed using the following terms: “long COVID”, “post-COVID-19”, “condition post-COVID-19”, “post-COVID-19 syndrome”, “COVID long haulers”, “long-term symptoms COVID-19”, “long-term COVID-19”, “COVID chronic”, “long infection COVID”, “post-acute COVID”, and “post-acute sequelae of SARS-CoV-2 infection (PASC)”. The Developer option within Word was added to search the Pro Word Cloud add-in and create the word cloud according to the search performed in the database and the frequency of the terms presented. The consultation and analysis were updated in June 2024. Source: PubMed—2024 [19].

**Figure 2 viruses-16-01256-f002:**
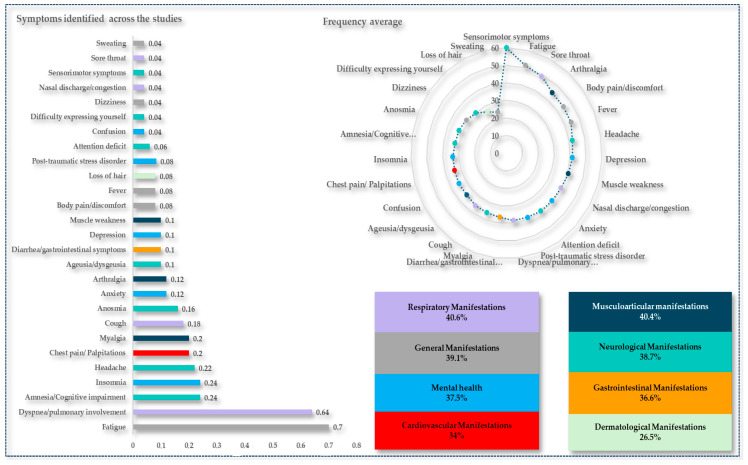
The frequencies of the main symptoms or sequelae were described and grouped into different organs and systems. We show the frequency in which the symptoms were mentioned in the studies, and the average frequency of the symptoms identified. Source: Authors.

**Table 1 viruses-16-01256-t001:** Observational studies that highlight LC syndrome, showing the prevalence (percentage of patients with the syndrome), most common symptoms, ordered by frequency above 20%, and duration of symptoms. The following databases were used: PubMed, MedRxiv, ScienceDirect, and Sage Journals.

Studies	Years	*n*	Ethnicity	Sex		Gravity	Time (Days)	Prevalence LC	Most Common Reported Symptoms
				Female	Male				
[107]	2021	172	Africa	65.7%	34.3%	Mild 87%, Moderate 9%, Severe 4%	240–300	65%	Fatigue (37.3%), Dyspnea (22%), Depression (22%)
[108]	2021	115	Africa	100%	0	-	45	77%	Sleep disturbance (63.5%), Fatigue (57.4%), Stress (56.5%), Sadness (47.8%), Cognitive dysfunction (25.2%), Recurrent falls (25.2%)
[109]	2023	524	Asia	40.5%	59.5%	Mild or moderate 96.4%, Severe 2.1%	73	8.2%	Fatigue (34.9%), Cough (27.9%)
[42]	2021	110	Europe	44%	56%	Hospitalized 100%	83	74%	Dyspnea (39%), Fatigue (39%), Insomnia (24%), Myalgia (22%)
[110]	2022	377	Europe	81.7%	18.3%	Hospitalized 100%	102	69%	Fatigue (39.5%), Dyspnea (28.9%), Myalgias (21.2%), Brain fog (20.2%)
[111]	2023	1802	North America	55.3%	44.7%	Mild 81%, Hospitalized 19%	240	55%	Cognitive impairment (51%), Change in lung function (44.9%)
[112]	2021	655	South America	64.7%	35.3%	Mild or moderate 92.4%, Severe 7.6%	76	21.8%	Olfactory dysfunction (53.8%), Taste deficit (68.3%)
[11]	2020	143	Europe	37%	63%	Hospitalized 100%	60	87%	Fatigue (53%), Dyspnea (43%), Arthralgias (27%), Chest pain (21.7%)
[113]	2021	150	Europe	56%	44%	Mild 100%	60	66%	Asthenia (40%), Dyspnea (36.7%), Anosmia (23%), Ageusia (23%)
[114]	2021	798	Africa	62.9%	37.1%	Mild 74.3%, Moderate or severe 25.7%	180	77.4%	Fatigue (42.2%), Shortness of breath (41.5%), Headaches (22.1%), Chest pain (20.3%)
[115]	2022	2333	Asia and Europe	49.2%	50.8%	Asymptomatic, Mild, Moderate 63%, Severe 37%	46	67.2%	Fatigue (38%), Dyspnea (30%), Myalgia (21.1%), Memory impairment (20.5%)
[3]	2021	3762	Europe and North America	78.9%	21.1%	Mild 91.6%, Severe 8.4%	210	55.9%	Fatigue (80%), Malaise (73.3%), Cognitive dysfunction (58.4%), Sensorimotor symptoms (55.7%), Headache (53.6%), Memory problems (51%), Insomnia (21%)
[116]	2021	119	Europe	38%	62%	Hospitalized 100%	60	87%	Fatigue (68%), Sleep disorders (57%), Dyspnea (32%), Post-traumatic stress disorder (25%), Anxiety (22%)
[117]	2021	201	Europe	71%	29%	Mild 81.6%, Severe 18.4%	140	99%	Fatigue (98%), Myalgia (86.7%), Dyspnea (87.1%), Headache (82.6%), Arthralgias (78.1%), Fever (75.1%), Cough (73. 6%), Chest pain (73.1%), Sore throat (71.1%), Diarrhea (59.2%), Pain (53.7%), Wheezing (48.3%), Muscle weakness (40.3%), Runny nose (33.8%)
[118]	2023	536	Europe	73%	27%	Mild 87%, Hospitalized 13%	360	62%	Fatigue (64%), Myalgia (35%), Shortness of breath (47%), Headache (34%), Chest pain (38%)
[119]	2022	1873	South Africa	51.3%	48.8%	Mild 29%, Moderate 29%, Severe 42%	120	66.7%	Fatigue (65.5%), headache (21.7%)
[120]	2023	1480	North America	75.1%	24.9%	Mild 98.53%, Hospitalized 1.47%	360	32.2%	Fatigue (48.3%), Dyspnea (22.9%), Confusion (22.7%), Headache (21.6%), Anosmia (20.6%), Ageusia (20.6%)
[121]	2023	1638	Africa	65.3%	34.7%	Mild 83%, Moderate 11.8%, Severe 4.8%	90	36.5%	Fatigue (24.6%), Mild depression (91.67%), Anxiety (93.25%)
[122]	2022	118	Africa	71%	29%	Mild 100%	120	47.5%	Asthenia (25.3%)
[123]	2020	120	Europe	37.5%	62.5%	Hospitalized 100%	110	55%	Fatigue (55%), Dyspnea (42%), Memory problems (34%), Sleep disorders (30.8%), Impaired concentration (28%), Hair loss (20%)
[124]	2021	430	Africa	63.7%	36.3%	Mild 83.3%%, Moderate 12%, Severe 4.7%	176	86%	Decreased daily activities (57.0%), Nervousness or hopelessness (53.3%), Sleeping troubles (50.9%); Cough (29.3%), Dyspnea (29.1%); Chest pain (32.6%); Anorexia (42.6%), Gastritis (32.3%); Myalgia (60.0%), Arthralgia (57.2%)
[26]	2021	100	Europe	46%	54%	Moderate 70%, Severe 30%	48	60–72%	Fatigue (72%), Dyspnea (65%), Stress (47%)
[37]	2021	1733	Asia	48%	52%	Hospitalized 100%	186	76%	Fatigue (63%), Muscle weakness (63%), Pain (27%), Dyspnea (26%), Insomnia (26%), Anxiety (23%), Depression (23%), Hair loss (22%)
[125]	2023	3700	South Africa	59%	41%	Mild 14%, Moderate or Severe 86%	180	38.5%	Fatigue (32.1%)
[126]	2022	538	Africa	54.1%	45.9	Mild 61.3%, Moderate 31%, Severe 7.7%	83	84.6%	Fatigue (59.1%), Sense of fever (46.5%), Anorexia (24.3%), Diarrhea (24.3%) Loss of taste (21.7%), Loss of smell (22.9%), Headache (21.4%), Cough (20.8), Dyspnea (21%)
[127]	2021	223	North America	53%	47%	Mild 71% Moderate or severe 29%	>90	28%	Chest Pain (24.2%), Fatigue/Weakness/Tiredness (50%), Hair Loss (35.5%), Anosmia (30%), Brain Fog/Cognitive Issues (22.6%), Anxiety (35%), Insomnia (22.6%)
[128]	2021	103	Europe	48%	52%	Hospitalized 100%	120	24%	Dyspnea (52%)
[129]	2020	76	Asia	72.4%	27.6%	Hospitalized 100%	120	42%	Chest pain (62%), Dyspnea (61%), Cough (60%), Fatigue (59%), Sputum (43%), Diarrhea (26%), Fever (20%)
[44]	2020	66	Asia	46.3%	53.7%	Mild 78.3%, Severe 21.7%	120	55%	Memory loss (28.3%), Myalgia (25%)
[130]	2023	7051	South America	74.1%	25.9%	Mild 100%	180	27.4%	Headache (53.4%), Myalgia (46.6%), Arthralgia (46.6%), Nasal congestion (45.1%), Fatigue (38%), Fever (36%), Dyspnea (35%), Cough (28%), Pain throat (27%)
[131]	2022	174	South Africa	62.1%	37.9%	Mild 98%, Moderate or Severe 2%	60	60.3%	Fatigue (34.5%), dyspnea (20.1%)
[132]	2023	1,913,234	Asia	50.6%	49.4%	Mild 100%	180–360	30%	Dyspnea (35.4%), Weakness (50.2%), Palpitations (22.1), Dizziness (42%), Arthralgia (39.6)
[133]	2021	277	Europe	47.3%	52.7%	Mild 34.3%, Severe 65.7%	77	51%	Fatigue (35%), Dyspnea (35%), Anosmia (38.5%), Dysgeusia (38.5%), Myalgias (20%), Arthralgia (20%)
[134]	2022	16,091	North America	62.6%	37.4%	Mild 100%	360	14.7%	Fatigue (52.2%), Confusion (45.7%), Loss of smell (43.7%), Brain fog (40.4%), Shortness of breath (39.7%), Headache (33.6%), Insomnia (30%), Anxiety (28.7%), Depression (23.3%), Dizziness (20.6%)
[27]	2021	180	Europe	54%	46%	Mild 95.6%, Severe 4.4%	125	55%	Fatigue (28.9%), Anosmia (27.2%)
[135]	2021	48	Europe	31.2%	68.8%	Hospitalized 100%	120	92%	Dyspnea (32.5%)
[24]	2022	96	Europe	55.2%	44.8%	Hospitalized 32.3%	360	77.1%	Reduced exercise capacity (56.3%), Fatigue (53.1%), Dyspnea (37.5%), Concentration problems (39.6%), Difficulty finding words (32.3%), Insomnia (26%)
[136]	2021	60	North America	32%	68%	Hospitalized 100%	120	58%	Dyspnea (20%), Cough (20%)
[137]	2021	145	Europe	43%	57%	Mild 25%, Severe 75%	103	41%	Dyspnea (36%), Pain (24%), Night sweats (24%), Insomnia (22%)
[10]	2021	4182	Europe	57%	43%	Mild 86.1%, Severe 13.9%	120	20.1%	Fatigue (97.7%), Headache (91.2%)
[30]	2021	128	Europe	54%	46%	Mild 44.5%, Severe 55.5%	75	62%	Fatigue (52.3%)
[39]	2021	22	Europe	27.3%	72.7%	Hospitalized 100%	120	55%	Dyspnea (48%)
[36]	2021	124	Europe	40%	60%	Mild 21.8%, Moderate 41%, Severe 37.1%	120	99%	Fatigue (69%), Functional impairment (64%), Cognitive impairments (36%)
[138]	2021	767	Europe	33%	67%	Hospitalized 100%	81	51.4%	Fatigue (51%), Dyspnea (51%), Post-traumatic stress (30.5%)
[139]	2020	78	North America	36%	64%	Hospitalized 100%	120	76%	Dyspnea (50%), Cough (23%)
[140]	2020	18	Europe	57.95	42.05%	Mild 33%, Severe 61%	85	78%	Attention deficits (50%, Concentration deficits (44.4%), Memory deficits (44.4%), Difficulty finding words (27.8%)
[141]	2022	62	South Africa	75.8%	24.2%	Mild 88.7%, Moderate 6.5%, Severe 4.8%	120	24.2%	Fatigue (42%), Anxiety (34%), Difficulty sleeping (31%), Brain fog (21%); Chest pain (24%); Muscle pain (21%)
[142]	2021	538	Asia	54.5%	45.5%	Hospitalized 100%	97	49.6%	Hair loss (28.6%), Fatigue (28.3%), Sweating (23.6%), Post-activity polypnea (21.4%)
[35]	2020	55	Asia	41.8%	58.2%	Mild 7.3%, Moderate 85.5%, Severe 7.3%	120	71%	Gastrointestinal symptoms (30.91%)
[143]	2022	302	Africa	58.7%	41.3%	Mild 100%	120	17.4%	Headache (25.9%), Cough (37.0%), Chest pain (22.2%)

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
