# Peer review of "The Silent Syndrome of Long COVID and Gaps in Scientific Knowledge: A Narrative Review"

_viruses, 2024, doi:10.3390/v16081256_

Round 1

Reviewer 1 Report

Comments and Suggestions for Authors

General Comments

·       This narrative review discussed the long COVID-19 syndrome in terms of pathophysiology, clinical pattern, risk factors and, genetic determinant.

·       First of all, it is difficult to refer to specific points across the manuscript given the lack of line numbering

·       In general, the manuscript is very long and the reader may get lost, as some aspects are reiterated several times. The various concepts should be summarized and presented in a more structured fashion, avoiding redundancies. For instance, sections “Clinical manifestations” and “Epidemiology” and “risk factors” cover overlapping aspects, e.g. symptoms or the increased risk in females. Moreover, the manuscript is a bit disjoint and lacks an overall flow.

·       The paper presents LC as a current problem, where it is a syndrome almost exclusively regarding the initial phase of the pandemic, when more virulent viral strains were circulating, vaccination not available yet and severe cases of COVID-19 were more prevalent. It is in fact expected that following an infectious disease there is a period of convalescence before recovery, and this period increases with the severity of the acute disease and hospital admission [recommended citation: PMID: 38140174]. However, the pandemic scenario changed dramatically at the end of 2021, when Omicron spread and population immunity against COVID-19 developed by either natural infection or vaccination. With Omicron the hospitalization rate dropped to <0.3% ( https://www.ecdc.europa.eu/en/covid-19/latest-evidence/clinical). Therefore, this review presented LC in an outdated manner, failing to stress that the pandemic is over, the syndrome is almost exclusively pertaining the past - i.e. the initial stage of the pandemic - and most patients developing the syndrome tended to return to a state of full health with time [recommended citation: PMID: 36676046].  Now COVID-19 and its sequelae is only a problem for high risk patients, at higher risk of developing severe disease.

·       Furthermore, LC dropped dramatically after reinfections, which almost exclusively surged during Omicron [recommended citations: PMID: 38140174; PMID: 37515237], although the rare reinfection observed during the first 1-2 pandemic years were reportedly more severe than the respective primary COVID-19 events.

·       Post COVID-19 symptoms (especially asthenia, but also other subjective conditions as myalgia or concentration deficit) should therefore be adequately screened. Moreover, patients with pre-existing psychological conditions (e.g. depression, anxiety) were more likely to report LC syndrome… thereby questioning the reliability of some of their post COVID19 sequelae (fatigue, concentration deficit, myalgia, other) [recommended citation: PMID: 36676046].

·       Furthermore, the problem with LC is not only the definition (time since COVID-19 diagnosis, time since disease healing, other) but also the denominator used to assess the impact of the syndrome. As mentioned, LC was more likely during the early pandemic waves, when more virulent strains were circulating and vaccination was still not available. With Omicron not only LC almost disappeared, at least in countries with high vaccination coverage, but also denominators enlarged and were mostly unknown, given the high proportion of asymptomatic COVID-19.

Specific comments

History and clinical definitions:

·        “As the pandemic progressed”… the manuscript has been circulating for a while? The pandemic is over by now…

·       The WHO proposed a clinical definition to unify several existing definitions”… “definition” repeated twice, please rephrase or find a synonym

·       Long COVID-19 was initially defined as a syndrome persisting 3+ months following acute disease, then the follow up was shortened to 4 weeks. Now restricting the follow up time to at least 2 weeks, as proposed by Raveedram seems a “desperate” attempt to defend a syndrome which is subjective and prone to “false positives”, especially since the main and most frequent symptom is asthenia, which is quite aspecific and could be subject to self-reported bias (false positive). Not to mention that patients might be influenced by social media.

Possible pathophysiological mechanism

Heart

·       Cardiovascular disease is the leading cause of death among both men and women, although epidemiological observations indicate that men are at lower risk than women. Estrogens are generally considered to have protective effects on the cardiovascular system because cardiovascular risk increases in women predominantly after menopause. Therefore, the onset of cardiovascular disease in women is delayed”… this paragraph is rather irrelevant for this article and can be removed.

·       This section could mention heart rate variability and vascular reactivity dysfunction at photoplethysmography-assessed cardiac autonomic system, which can be mitigated by slow breathing, a manoeuvre enhancing vagal response and baroreflex sensitivity, suggesting that repeated sessions of slow breathing exercise may play an important role in restoring vascular impairment in subjects with long-term cardiovascular effect of LC (https://doi.org/10.1016/j.amjmed.2024.05.021).

Other organs and systems

·       “are a risk factor for the development of functional gastrointestinal”… change to plural (“factors”)

·       What about work ability (measured i.e. by WAI (recommended citation: PMID: 36676046].

Inflammatory and immune response

·       .. “and leading to exudate, edema, and fibrosis occur”, this sentence needs rephrasing

·       SARS-CoV-2 can cause antigen-presenting cells to present antigens” repetition of “present

·       “The infection and destruction of lung cells trigger a local immune response through the recruitment of macrophages and monocytes that respond to infection by promoting the release of cytokines. On the other hand, in a smaller number of patients, there is an atypical immune response that triggers substantial activation of cytokines (cytokine storm), causing damage to infected cells and severe lung disease”. …The two scenarios seem identical, as both end up with the release of cytokines…

Section: Clinical Manifestations

·       Section “Clinical Manifestations”: the concept of variability is reiterated multiple time, e.g. the following sentence “Patients develop LC with a variety of symptoms and signs of varying degrees and durations” is redundant and can be removed

·       symptom chronification and potential predictors of chronification”… “chronification” is repeated twice; moreover “chronicization” is a better term

·       The rate of symptom resolution is reportedly even higher than 45%. For instance in a population based study conducted during the early pandemic stage 75% patients returned to their baseline status preceding COVID-19 after a median time of 15 months since COVID-19 diagnosis [PMID: 36676046].

·       Regarding the predictors of chronicity, the results showed an independent association between difficulty concentrating at 7 months and the chronification of symptoms”… this sentence does not make sense, since concentration deficit cannot be a predictor of chronicization, since it is a persisting symptom (hence a mark of LC rather than a predictor). Predictors of LC may be age, sex, pre-existing conditions, severity of COVID-19, etc.

·       Searching criteria to collect the available literature: “The following characteristics were extracted from the studies: first author, year, country, sample, sex, severity, duration of symptoms, prevalence (percentage of patients with the syndrome), and most common symptoms ordered by frequency above 20% (Table 1)” … what year were considered? It seems some relevant publications have been neglected.

·       Table 1 needs to report also the year of publication of each article

·       Figure 2 shows the main symptoms cited in the studies presented in Table 1 according to the frequency of citations”… replace “cited with “self-reported

·       Approximately 12% to 15% of patients who have mild symptoms still had symptoms for up to 8 months”… “symptoms” repeated twice

·       In addition, the number of symptoms in patients with LC may be associated with the increased number of symptoms during the acute phase”, awkward expression with repetition of the word “symptoms”

·       “… survival of COVID-19 between men and women infected with SARS-CoV-2”, replace “men” with “males” and “women” with “females”, the same applies across the entire manuscript

·       Evidence suggests that the severity and mortality of COVID-19 are greater in men than in womenwhereas women seem to have a greater risk of reinfection and the development of LC” … SARS-CoV-2 infections can be asymptomatic/mild, hence they can be detected only by systematic screening schedule as done in health care workers. Moreover, HCWs are also more reliable to assess the impact of self-reported symptoms. Whilst LC is more frequent in females, and female HCWs were more likely to be SARS-CoV-2 re-infected, reinfections were all mild or asymptomatic, also because they occurred almost exclusively during the Omicron time [recommended citation: PMID: 37515237]. Therefore, reinfections in females have little to do with LC, since the risk of LC typically increases with the severity of acute disease and duration of viral shedding time [PMID: 38140174]. Reinfections during the first pandemic year were very few but reportedly more severe (according to a bunch of case reports). However, not during Omicron, when reinfections surged but were by far mild/asymptomatic.

·       Nevertheless, most people have good functional recovery during 1 year of follow-up”…. Most patients returned to a state of full health with time

Risk factors

·       LC has become a complex and heterogeneous syndrome that may have multiple triggering factors. According to Sudre et al. (2021), these factors include disease severity, age (over 50 years), sex (female), and pre-existing comorbidities, such as asthma or respiratory disease, obesity, and high body mass index

·       Women are more affected by LC than men. According to the WHO, one in three women and one in five men are likely to develop LC [139]. This situation may be due to immunological and hormonal differences between the sexes”, this is a repetition, as this point was already raised in the previous section

·       Regarding mental health, pre-pandemic psychological distress was associated with a greater risk of LC outcomes for more than 12 weeks”, this is a major risk-factor. Please also cite findings from the following population based study during the early pandemic period, where pre-existing depression was a risk factor for LC [PMID: 36676046].

·       Another recent case-control study with 7,051 health professionals showed that having more than one SARS-CoV-2 infection, especially those infected with the Delta variant or the Omicron variant, was an important risk factor for LC”… this is a rather questionable point; the authors cited the pre-print, however the article was eventually published.… 2+ infections were considered before developing LC?

·       A study in the United Kingdom involving more than 28,000 patients showed a 12.8% reduction in the chances of developing LC after the first dose of the vaccine and an additional 8.8% reduction after the second dose”, it makes sense, as the first or two doses were administered during 2021, when more virulent variants (Alpha, Delta) were circulating. By contrast, 3+ doses were delivered during Omicron time, when the risk of severe disease was significantly lower [PMID: 38140174]. This confirms that LC was mainly an issue during the early pandemic waves, not during Omicron.

·       prolonged viral shedding time (VST) was not discussed among risk factors [recommended citation: PMID: 38140174; https://doi.org/10.3390/pathogens13050388]

Association with cellular, molecular, and genetic factors

·       The findings showed that the proportions of symptoms in the acute phase of the disease were higher among patients who progressed to the clinical picture of LC.”, this point reinforces the concept that LC much depends on the severity of acute disease.

·       Changes in proinflammatory biomarkers have been reported in LC. Two months after hospitalization for COVID-19, lung lesions were associated with elevated systemic inflammatory biomarkers, such as D-dimer, IL-6, and CRP. Multivariate regression showed that the IL-6 level at admission was an independent factor associated with persistent lung injury [161]. Another study found that 3 months after hospital discharge, COVID-19 survivors had high levels of blood urea nitrogen and D-dimer, which are considered risk factors for pulmonary dysfunction [35]. Among LC patients followed for 3 months, Liang et al. (2020) reported that lymphopenia correlated with chest tightness and heart palpitations, while elevated troponin-I correlated with fatigue ”… this paragraph somehow repeats point already pinned down in section “Inflammation and immune response”

·       When discussing heart rate variability I recommend to cited also findings from this recent observational study: https://doi.org/10.1016/j.amjmed.2024.05.021 Reference: AJM 17573

Comments on the Quality of English Language

Acceptable, although it can be improved

Author Response

Reviewer 1

General Comments

This narrative review discussed the long COVID-19 syndrome in terms of pathophysiology, clinical pattern, risk factors and, genetic determinant.

First of all, it is difficult to refer to specific points across the manuscript given the lack of line numbering

R: Sorry for this. The Journal has a template with line numbering, and we checked and are sure we sent this template correctly structures and with line numbering. Moreover, the used line number references in their reviews. So, unfortunately this should be some problem with the files you received or accessed from the journal. Again, we really sorry for this inconvenience.

  • In general, the manuscript is very long and the reader may get lost, as some aspects are reiterated several times. The various concepts should be summarized and presented in a more structured fashion, avoiding redundancies. For instance, sections “Clinical manifestations” and “Epidemiology” and “risk factors” cover overlapping aspects, e.g. symptoms or the increased risk in females. Moreover, the manuscript is a bit disjoint and lacks an overall flow.

R: Long Covid is the major theme in Covid literature. We tried to accommodate information of hundred of studies in our narrative. Your perception about style and redundances are very useful. We tried to improve the manuscript keeping your comments in mind.

The paper presents LC as a current problem, where it is a syndrome almost exclusively regarding the initial phase of the pandemic, when more virulent viral strains were circulating, vaccination not available yet and severe cases of COVID-19 were more prevalent. It is in fact expected that following an infectious disease there is a period of convalescence before recovery, and this period increases with the severity of the acute disease and hospital admission [recommended citation: PMID: 38140174]. However, the pandemic scenario changed dramatically at the end of 2021, when Omicron spread and population immunity against COVID-19 developed by either natural infection or vaccination. With Omicron the hospitalization rate dropped to <0.3% ( https://www.ecdc.europa.eu/en/covid-19/latest-evidence/clinical). Therefore, this review presented LC in an outdated manner, failing to stress that the pandemic is over, the syndrome is almost exclusively pertaining the past - i.e. the initial stage of the pandemic - and most patients developing the syndrome tended to return to a state of full health with time [recommended citation: PMID: 36676046].  Now COVID-19 and its sequelae is only a problem for high risk patients, at higher risk of developing severe disease.

  • Furthermore, LC dropped dramatically after reinfections, which almost exclusively surged during Omicron [recommended citations: PMID: 38140174; PMID: 37515237], although the rare reinfection observed during the first 1-2 pandemic years were reportedly more severe than the respective primary COVID-19 events.
  • Post COVID-19 symptoms (especially asthenia, but also other subjective conditions as myalgia or concentration deficit) should therefore be adequately screened. Moreover, patients with pre-existing psychological conditions (e.g. depression, anxiety) were more likely to report LC syndrome… thereby questioning the reliability of some of their post COVID19 sequelae (fatigue, concentration deficit, myalgia, other) [recommended citation: PMID: 36676046].
  • Furthermore, the problem with LC is not only the definition (time since COVID-19 diagnosis, time since disease healing, other) but also the denominator used to assess the impact of the syndrome. As mentioned, LC was more likely during the early pandemic waves, when more virulent strains were circulating and vaccination was still not available. With Omicron not only LC almost disappeared, at least in countries with high vaccination coverage, but also denominators enlarged and were mostly unknown, given the high proportion of asymptomatic COVID-19.

R: Your general comments were approached during our responses to your specific comments, along with the inclusion of the many studies you suggested. We hope we addressed all of them. Thank you very much for your detailed work improving our manuscript.

Specific comments

History and clinical definitions:

  • “As the pandemic progressed”… the manuscript has been circulating for a while? The pandemic is over by now…

R: Thank you. We changed it and now is: During the first year of the pandemics

  • The WHO proposed a clinical definition to unify several existing definitions”… “definition” repeated twice, please rephrase or find a synonym

R: We changed to: The WHO proposed a clinical nomenclature to unify several existing definitions

  • Long COVID-19 was initially defined as a syndrome persisting 3+ months following acute disease, then the follow up was shortened to 4 weeks. Now restricting the follow up time to at least 2 weeks, as proposed by Raveedram seems a “desperate” attempt to defend a syndrome which is subjective and prone to “false positives”, especially since the main and most frequent symptom is asthenia, which is quite a specific and could be subject to self-reported bias (false positive). Not to mention that patients might be influenced by social media.

R: In the section History and case definition we presented the several propositions for clinical diagnosis and classification. We just exposed them, from NICE to Raveedram. Actually, our objective was to display the heterogeneity and difficulties existing in literature. We did not make judgments about “desperation” or not. Indeed, our last paragraph in this section initiates with the sentence “Notably, there is currently no definitive diagnosis of LC.” When we found this heterogeneous diagnostic scenario in literature and the huge number of papers about long covid we became clear that Reviews like ours are needed, exposing these issues and promoting efforts to unify and solve them.

Possible pathophysiological mechanism

Heart

  • Cardiovascular disease is the leading cause of death among both men and women, although epidemiological observations indicate that men are at lower risk than women. Estrogens are generally considered to have protective effects on the cardiovascular system because cardiovascular risk increases in women predominantly after menopause. Therefore, the onset of cardiovascular disease in women is delayed”… this paragraph is rather irrelevant for this article and can be removed.

R: Thank you, we agree. The paragraph was excluded 

  • This section could mention heart rate variability and vascular reactivity dysfunction at photoplethysmography-assessed cardiac autonomic system, which can be mitigated by slow breathing, a manoeuvre enhancing vagal response and baroreflex sensitivity, suggesting that repeated sessions of slow breathing exercise may play an important role in restoring vascular impairment in subjects with long-term cardiovascular effect of LC (https://doi.org/10.1016/j.amjmed.2024.05.021).

R: Thank you for this valuable suggestion, we included in the manuscript

Other organs and systems

  • “are a risk factor for the development of functional gastrointestinal”… change to plural (“factors”)

R: Thank you, done

  • What about work ability (measured i.e. by WAI (recommended citation: PMID: 36676046].

R: We included at the end of the section “Clinical Manifestations”

Inflammatory and immune response

  • .. “and leading to exudate, edema, and fibrosis occur”, this sentence needs rephrasing

R: The sentence was rewrote.

  • SARS-CoV-2 can cause antigen-presenting cells to present antigens” repetition of “present

R: The sentence was rewrote

  • “The infection and destruction of lung cells trigger a local immune response through the recruitment of macrophages and monocytes that respond to infection by promoting the release of cytokines. On the other hand, in a smaller number of patients, there is an atypical immune response that triggers substantial activation of cytokines (cytokine storm), causing damage to infected cells and severe lung disease”. …The two scenarios seem identical, as both end up with the release of cytokines…

R: We could not see the similarity. The first initiates with the infection that damages lung cells that attracts and stimulates macrophages and monocytes to respond and release cytokines. The second initiates with cytokine storm that lead to cell damage and lung disease. In the firs the lung damage is the start point and in the second is the consequence. Cytokine storm is not only an elevation in the production of cytokines as well as not all elevation in cytokine levels becomes a cytokine storm. However, we rewrote this text and hope it becomes clearer.

Section: Clinical Manifestations

  • Section “Clinical Manifestations”: the concept of variability is reiterated multiple time, e.g. the following sentence “Patients develop LC with a variety of symptoms and signs of varying degrees and durations” is redundant and can be removed

R: We rewrote, thank you 

  • symptom chronification and potential predictors of chronification”… “chronification” is repeated twice; moreover “chronicization” is a better term

R: Thank you, we rewrote the sentence.

  • The rate of symptom resolution is reportedly even higher than 45%. For instance in a population based study conducted during the early pandemic stage 75% patients returned to their baseline status preceding COVID-19 after a median time of 15 months since COVID-19 diagnosis [PMID: 36676046].

R: We improved the discussion of this theme and included this study.

  • Regarding the predictors of chronicity, the results showed an independent association between difficulty concentrating at 7 months and the chronification of symptoms”… this sentence does not make sense, since concentration deficit cannot be a predictor of chronicization, since it is a persisting symptom (hence a mark of LC rather than a predictor). Predictors of LC may be age, sex, pre-existing conditions, severity of COVID-19, etc.

R: You are right, we expressed the ideas wrongly. The text was corrected

  • Searching criteria to collect the available literature: “The following characteristics were extracted from the studies: first author, year, country, sample, sex, severity, duration of symptoms, prevalence (percentage of patients with the syndrome), and most common symptoms ordered by frequency above 20% (Table 1)” … what year were considered? It seems some relevant publications have been neglected.

R: This issue was also pointed by another Reviewer. We included a paragraph in Introduction about criteria of literature surveying. About the Table 1, we included more studies following the most recent systematic reviews. Thus, now we have 50 prevalence studies. Now we have more representativity worldwide. The Figure 2, derivated from Table 1, was also updated accordingly.

  • Table 1 needs to report also the year of publication of each article

R: We corrected it. Thank you 

  • Figure 2 shows the main symptoms cited in the studies presented in Table 1 according to the frequency of citations”… replace “citedwith “self-reported

R: We corrected

  • Approximately 12% to 15% of patients who have mild symptoms still had symptoms for up to 8 months”… “symptoms” repeated twice

R: Rewrote, thank you.

  • In addition, the number of symptoms in patients with LC may be associated with the increased number of symptoms during the acute phase”, awkward expression with repetition of the word “symptoms”R: The sentence was improved.
  • “… survival of COVID-19 between men and women infected with SARS-CoV-2”, replace “men” with “males” and “women” with “females”, the same applies across the entire manuscript

R: Thank you, we replaced all 

  • Evidence suggests that the severity and mortality of COVID-19 are greater in men than in womenwhereas women seem to have a greater risk of reinfection and the development of LC” … SARS-CoV-2 infections can be asymptomatic/mild, hence they can be detected only by systematic screening schedule as done in health care workers. Moreover, HCWs are also more reliable to assess the impact of self-reported symptoms. Whilst LC is more frequent in females, and female HCWs were more likely to be SARS-CoV-2 re-infected, reinfections were all mild or asymptomatic, also because they occurred almost exclusively during the Omicron time [recommended citation: PMID: 37515237]. Therefore, reinfections in females have little to do with LC, since the risk of LC typically increases with the severity of acute disease and duration of viral shedding time [PMID: 38140174]. Reinfections during the first pandemic year were very few but reportedly more severe (according to a bunch of case reports). However, not during Omicron, when reinfections surged but were by far mild/asymptomatic.

R: We included your comments about these points in an additional paragraph.

  • Nevertheless, most people have good functional recovery during 1 year of follow-up”…. Most patients returned to a state of full health with time

R: We improved the text where resolution time is comparatively discussed. 

Risk factors

  • LC has become a complex and heterogeneous syndrome that may have multiple triggering factors. According to Sudre et al. (2021), these factors include disease severity, age (over 50 years), sex (female), and pre-existing comorbidities, such as asthma or respiratory disease, obesity, and high body mass index
  • Women are more affected by LC than men. According to the WHO, one in three women and one in five men are likely to develop LC [139]. This situation may be due to immunological and hormonal differences between the sexes”, this is a repetition, as this point was already raised in the previous section

R: We recall this point because we included more information suggested by you. The data from WHO was not previously cited.

  • Regarding mental health, pre-pandemic psychological distress was associated with a greater risk of LC outcomes for more than 12 weeks”, this is a major risk-factor. Please also cite findings from the following population based study during the early pandemic period, where pre-existing depression was a risk factor for LC [PMID: 36676046].

R: We included a paragraph after Figure 2 citing Sansone´s papers along with others

  • Another recent case-control study with 7,051 health professionals showed that having more than one SARS-CoV-2 infection, especially those infected with the Delta variant or the Omicron variant, was an important risk factor for LC”… this is a rather questionable point; the authors cited the pre-print, however the article was eventually published.… 2+ infections were considered before developing LC?

R: We checked the published study, and some slight changes were made by the authors. We changed them accordingly in our manuscript.

  • A study in the United Kingdom involving more than 28,000 patients showed a 12.8% reduction in the chances of developing LC after the first dose of the vaccine and an additional 8.8% reduction after the second dose”, it makes sense, as the first or two doses were administered during 2021, when more virulent variants (Alpha, Delta) were circulating. By contrast, 3+ doses were delivered during Omicron time, when the risk of severe disease was significantly lower [PMID: 38140174]. This confirms that LC was mainly an issue during the early pandemic waves, not during Omicron.

R: We reorganized the paragraphs about reinfections, vaccination and viral strains and now they are close and show more clearly the agreements and controversies. We believe the text is now more comprehensive about these themes, thanks your contributions.

  • prolonged viral shedding time (VST) was not discussed among risk factors [recommended citation: PMID: 38140174; https://doi.org/10.3390/pathogens13050388]

R: We included Sansone´s reference and VST in a new paragraph, as previously pointed. 

Association with cellular, molecular, and genetic factors

  • The findings showed that the proportions of symptoms in the acute phase of the disease were higher among patients who progressed to the clinical picture of LC.”, this point reinforces the concept that LC much depends on the severity of acute disease.

R: We are aware these results form our ow study. But this can be tricky because the proportion of symptoms are not always higher among severe patients. The multiplicity of symptoms during acute infection includes symptoms more frequent in mild disease, like taste and olfactory impairments as well as symptoms more typical of severe forms, like the respiratory ones. Hence, we opted to not make this kind of link.

  • Changes in proinflammatory biomarkers have been reported in LC. Two months after hospitalization for COVID-19, lung lesions were associated with elevated systemic inflammatory biomarkers, such as D-dimer, IL-6, and CRP. Multivariate regression showed that the IL-6 level at admission was an independent factor associated with persistent lung injury [161]. Another study found that 3 months after hospital discharge, COVID-19 survivors had high levels of blood urea nitrogen and D-dimer, which are considered risk factors for pulmonary dysfunction [35]. Among LC patients followed for 3 months, Liang et al. (2020) reported that lymphopenia correlated with chest tightness and heart palpitations, while elevated troponin-I correlated with fatigue ”… this paragraph somehow repeats point already pinned down in section “Inflammation and immune response”

R: Despite we discuss these points in terms of mechanism during the section Inflammation and Immune Response. We did not approach them as risk factors. In this paragraph from Risk factors section, we pinpointed the role of them as risk factors an included references not cited in the Inflammation section.

  • When discussing heart rate variability I recommend to cited also findings from this recent observational study: https://doi.org/10.1016/j.amjmed.2024.05.021 Reference: AJM 17573

R: We already have agreed with this suggestion you gave previously and included this information. Thank you.

Reviewer 2 Report

Comments and Suggestions for Authors

Thank you for giving me an opportunity to check your peer review titled 'The silent syndrome of long COVID and gaps in scientific knowledge: A narrative review'.  It's an acceptable article about COVID-19 as a peer review about COVID-19. You summarized many knowledge in this article in previous articles, but some of them were not latest. Many physicians are known almost of them I think. But, in the view of immunology and cytokines field, it's very interesting for me and many physicians. There are still many unclear knowledge in symptoms and sequelae of COVID-19. 

Your article is very important for us to update the novel knowledge for COVID-19. So, my opinion is "accept" for publishing to this journal.

Author Response

Reviewer 2

Thank you for giving me an opportunity to check your peer review titled 'The silent syndrome of long COVID and gaps in scientific knowledge: A narrative review'.  It's an acceptable article about COVID-19 as a peer review about COVID-19. You summarized many knowledge in this article in previous articles, but some of them were not latest. Many physicians are known almost of them I think. But, in the view of immunology and cytokines field, it's very interesting for me and many physicians. There are still many unclear knowledge in symptoms and sequelae of COVID-19. 

Your article is very important for us to update the novel knowledge for COVID-19. So, my opinion is "accept" for publishing to this journal.

R: We are grateful for your motivating commentaries.

Reviewer 3 Report

Comments and Suggestions for Authors

Silva et al discussed the present understanding of the etiology of long COVID. They discussed the history, reason and consequences of long COVID   in lung, heart, brain and other organs. They should improve the manuscript based on the following comments.

1.       In introduction, would it be NICE or NIHCE?

2.       The term “PASC (post-acute sequelae of SARS-CoV-2 infection (PASC)” is not mentioned in text or in figure 1 as a synonym for long COVID.

3.       Authors should discuss more about “new symptoms appear”- a probable explanation of how new symptoms can occur?

4.       ARDS should be expanded.

5.      Epidemiological observations indicate that men are at lower risk than women” need reference

6.       The role of ‘microbiota’ in developing long COVID is not discussed.

7.       The role of serotonin storage abnormalities should be included for neuronal malfunction; doi: 10.1016/j.cell.2023.09.013

8.       Mitochondrial dysfunction in long COVID should be discussed; doi: 10.1038/s41467-023-44432-3

9.       The activity of abzyme in long covid should be mentioned 10.1128/mbio.00541-24.

10.   English should be moderately edited.

Comments on the Quality of English Language

Moderate English editing is required.

Author Response

Reviewer 3

Silva et al discussed the present understanding of the etiology of long COVID. They discussed the history, reason and consequences of long COVID in lung, heart, brain and other organs. They should improve the manuscript based on the following comments.

  1. In introduction, would it be NICE or NIHCE?

R: Sounds strange but NICE is right, according to: https://www.nice.org.uk/

  1. The term “PASC (post-acute sequelae of SARS-CoV-2 infection(PASC)” is not mentioned in text or in figure 1 as a synonym for long COVID.

R: We included it in the introduction and Figure1. We agree that this information is important. Thank you very much.

  1. Authors should discuss more about “new symptoms appear”- a probable explanation of how new symptoms can occur?

R: Unfortunately, the literature does not explore this issue! There are no explanation how new symptoms, not observed during the acute disease, appears. The physiopathology of LC is still unknown. Thus, this mechanism remains to be clarified. We made explicit in manuscript´s text that the mechanism is till now unknown.

  1. ARDS should be expanded.

R: Thank you we expanded it

  1. “Epidemiological observations indicate that men are at lower risk than women” need reference.

R: Because another Reviewer pointed that the relevance of this paragraph is not so high we opted to exclude the entire Paragraph

  1. The role of ‘microbiota’ in developing long COVID is not discussed.

R: We included this in the Other organs and systems section. Thank you very much.

  1. The role of serotonin storage abnormalities should be included for neuronal malfunction; doi: 10.1016/j.cell.2023.09.013

R: We included this in the Other organs and systems section. Thank you very much.

  1. Mitochondrial dysfunction in long COVID should be discussed; doi: 10.1038/s41467-023-44432-3

R: We included this in the Clinical manifestations section. Thank you very much.

  1. The activity of abzyme in long covid should be mentioned 1128/mbio.00541-24.

R: We included it in the Heart. Thank you very much.

Reviewer 4 Report

Comments and Suggestions for Authors

While the paper provides valuable insights, several key elements need improvement for it to be more impactful and scientifically rigorous.

Clear Focus: The review lacks a clear focus. It would benefit from a more defined scope and specific research questions that guide the narrative. This would help readers understand the central theme and objectives of the review.

Knowledge Gap: The manuscript does not clearly identify the existing knowledge gap. Highlighting what is unknown or what gaps the review aims to fill is crucial for emphasizing the importance and relevance of the study.

Reproducibility: As with many narrative reviews, the research methodology is not reproducible. It would be beneficial to outline the search strategies and selection criteria in more detail, even if they are less structured than systematic reviews. This would enhance the transparency and reliability of the findings.

Strengths and Limitations: The manuscript is missing a section that discusses the strengths and limitations of the review. Including this section is essential for providing a balanced perspective on the findings and acknowledging the inherent constraints of a narrative review.

There is a lack of detail regarding the hypercoagulability mentioned. The text should include specific markers and evidence from recent studies that link hypercoagulability to LC. For example, the discussion could benefit from including more detailed information on D-dimer levels and their implications.

Overall, the manuscript is a good paper with potential. By addressing these points, the authors can significantly improve its clarity, relevance, and scientific contribution.

Author Response

Reviewer 4

While the paper provides valuable insights, several key elements need improvement for it to be more impactful and scientifically rigorous.

Clear Focus: The review lacks a clear focus. It would benefit from a more defined scope and specific research questions that guide the narrative. This would help readers understand the central theme and objectives of the review. Knowledge Gap: The manuscript does not clearly identify the existing knowledge gap. Highlighting what is unknown or what gaps the review aims to fill is crucial for emphasizing the importance and relevance of the study.

R: Most of the Reviews, Narrative or Systematic, are indeed clearly focused and we agree that this helps a lot by organizing the information. However, during the exploratory reading of published papers, including reviews, we feel a lack of recent general Reviews, covering the major themes in an integrated and articulated way. Thus, we opted by a Narrative Review covering LC that would allow to detect what is more consolidated about LC and, by contrast, what constitutes a gap or controversy. We agree that we need to strength gaps and controversies and improved it during the text. Moreover, at the end of the manuscript, in the Conclusions and Future Prospects section we presented several knowledge gaps that we could identify. We hope it gives useful directions to future studies.

Reproducibility: As with many narrative reviews, the research methodology is not reproducible. It would be beneficial to outline the search strategies and selection criteria in more detail, even if they are less structured than systematic reviews. This would enhance the transparency and reliability of the findings.

 R: We introduced a new paragraph at the end of Introduction section addressing this issue.

Strengths and Limitations: The manuscript is missing a section that discusses the strengths and limitations of the review. Including this section is essential for providing a balanced perspective on the findings and acknowledging the inherent constraints of a narrative review.

R: We agree. We added Strengths and Limitations at the Conclusions and Future Prospects section

There is a lack of detail regarding the hypercoagulability mentioned. The text should include specific markers and evidence from recent studies that link hypercoagulability to LC. For example, the discussion could benefit from including more detailed information on D-dimer levels and their implications.

R: R: We included it in the Thrombotic events section. Thank you very much. 

Overall, the manuscript is a good paper with potential. By addressing these points, the authors can significantly improve its clarity, relevance, and scientific contribution.

R: Thank you very much.

Round 2

Reviewer 1 Report

Comments and Suggestions for Authors

The authors tried to address peer review recommendations.

However, the manuscript maintained its original structure as an hectic list of studies on LC.

Furthermore, the manuscript needs careful screening for the English used, as it if often confusing and difficult to read. Here are some examples of parts that need to be rephrased.

- Line 274: "factors" should be changed to "risk factors"

- Reference 59: the authors list mixed surnames with first names.

Correct autors's list by surnames is:

Mauro M, Cegolon L, Bestiaco N, Zuian E, Larese Filon F.

- Likeiwse (line 257) citation should be Mauro et al (2024)

Lines 257-259: to be rephrased as follows: "Finally, Mauro et al. (2024) reported decreased heart rate variability at rest, measured by finger-photoplethysmography, among 58 HCWs with LC in Trieste Universty hospital -  all but one previously develpoing mild COVID-19 - assessed after a median time of 419.5 days since COVID-19 diagnosis. Slow paced breathing was able to enhance vagal response and restore baroflex modulation and vascular reactiviy in HCWs with LC [59]

Line 278-280: this sentence is awkard and needs to be rephrased

Line 283-285: "With infection with many variants of SARS-CoV-2 (e.g. in the case of the omicron variant), D-dimer levels increase considerably, and a lot of clotting occurs, which can be reflected in LC"... confusing sentence, to be rephrased.

Line 310- 322: this entire paragraph does not read well. For instance, lines 320-22 are badly written and need ot be rephrased. 

Lines 339-344: However, there is a recent study which, ...  decline in platelet serotonin is expected"... this sentence is badly written adn confusing and needs rephrasing

Line 478: "Impairment"... impairment of what"

Line 480-81: "Patients develop LC with a several of symptoms and signs of varying intensity degrees and durations"... bad wording, moreover a reference is missing at the end of the sentence.

Line 481-82: "Logistic regression models were used to evaluate the associations between symptom chronification and their potential predictors of chronification".. where was logistic regression used

Line 487: "with a positive association"... with a positive associatin with what?

Line 495: "demonstrated"... change to "reported"

Line 574-77: "In addition, there is an association between LC and the number of symptoms during the acute phase of infection and with the number of comorbidities among non-hospitalized patients"... this is confusing, what is assocated with what? Maybe (to keep it simple) "Among non-hospitalized patients the risk of LC increased with number of symptoms during the acute disease and with pre-existing comorbidities?

Author Response

Reviewer_1,_second_round

The authors tried to address peer review recommendations.

However, the manuscript maintained its original structure as an hectic list of studies on LC.

Furthermore, the manuscript needs careful screening for the English used, as it if often confusing and difficult to read. Here are some examples of parts that need to be rephrased.

R: Thank you we are reviewing the English, and your suggestions were helpful and welcome. This kind of review we are writing is very difficult because there are controversies in the literature and several uncertain issues. So, if we approach a point is necessary to show the many views about it. Thus, it is difficult to avoid, sometimes, a listing of authors and studies. This scenario speaks favorably to the need for reviews like this because gives an overview, pinpointing the polemic or controversies. Thank you again for all your efforts to improve our work.

- Line 274: "factors" should be changed to "risk factors"

R: Thank you. Changed.

- Reference 59: the authors list mixed surnames with first names.

Correct autors's list by surnames is:

Mauro M, Cegolon L, Bestiaco N, Zuian E, Larese Filon F.

R: Thank you. It was a Mendeley bug that escaped us unseen.

- Likeiwse (line 257) citation should be Mauro et al (2024)

R: Thank you. Adjusted.

Lines 257-259: to be rephrased as follows: "Finally, Mauro et al. (2024) reported decreased heart rate variability at rest, measured by finger-photoplethysmography, among 58 HCWs with LC in Trieste Universty hospital -  all but one previously develpoing mild COVID-19 - assessed after a median time of 419.5 days since COVID-19 diagnosis. Slow paced breathing was able to enhance vagal response and restore baroflex modulation and vascular reactiviy in HCWs with LC [59]

R: Replaced.

Line 278-280: this sentence is awkard and needs to be rephrased

R: Thank you, we rewrote.

 Line 283-285: "With infection with many variants of SARS-CoV-2 (e.g. in the case of the omicron variant), D-dimer levels increase considerably, and a lot of clotting occurs, which can be reflected in LC"... confusing sentence, to be rephrased.

R: Checked these paragraphs and rewrote these sentences.

Line 310- 322: this entire paragraph does not read well. For instance, lines 320-22 are badly written and need ot be rephrased. 

R: Thank you, we rewrote.

Lines 339-344: However, there is a recent study which, ...  decline in platelet serotonin is expected"... this sentence is badly written adn confusing and needs rephrasing

R: Thank you, we rewrote.

Line 478: "Impairment"... impairment of what"

R: Thank you, we rewrote.

Line 480-81: "Patients develop LC with a several of symptoms and signs of varying intensity degrees and durations"... bad wording, moreover a reference is missing at the end of the sentence.

R: We improved it.

Line 481-82: "Logistic regression models were used to evaluate the associations between symptom chronification and their potential predictors of chronification".. where was logistic regression used

R: We improved it.

Line 487: "with a positive association"... with a positive associatin with what?

R: Corrected.

Line 495: "demonstrated"... change to "reported"

R: Replaced.

Line 574-77: "In addition, there is an association between LC and the number of symptoms during the acute phase of infection and with the number of comorbidities among non-hospitalized patients"... this is confusing, what is assocated with what? Maybe (to keep it simple) "Among non-hospitalized patients the risk of LC increased with number of symptoms during the acute disease and with pre-existing comorbidities?

R: Your interpretation is correct. Thank you. We replace our sentence with yours.

Reviewer 3 Report

Comments and Suggestions for Authors

The paper is much improved 

Comments on the Quality of English Language

Still some grammatical mistakes are present and can be corrected

Author Response

The paper is much improved 

Comments on the Quality of English Language

Still some grammatical mistakes are present and can be corrected

R: We are grateful for your contributions. We will review the language editing.

Reviewer 4 Report

Comments and Suggestions for Authors

All comments have been adequately addressed.

Author Response

Comments and Suggestions for Authors

All comments have been adequately addressed.

R: We are grateful for your contributions.